# Youth Positive Mental Health Concepts and Definitions: A Systematic Review and Qualitative Synthesis

**DOI:** 10.3390/ijerph191811506

**Published:** 2022-09-13

**Authors:** Janhavi Ajit Vaingankar, Mythily Subramaniam, Esmond Seow, Sherilyn Chang, Rajeswari Sambasivam, Nan Luo, Swapna Verma, Siow Ann Chong, Rob M. van Dam

**Affiliations:** 1Research Division, Institute of Mental Health, 10, Buangkok View, Singapore 539747, Singapore; 2Saw Swee Hock School of Public Health, National University of Singapore, Singapore 117549, Singapore; 3Yong Loo Lin School of Medicine, National University of Singapore, Singapore 119228, Singapore; 4Department of Early Psychosis, Institute of Mental Health, 10, Buangkok View, Singapore 539747, Singapore; 5Duke-NUS Medical School, 8 College Road, Singapore 169857, Singapore

**Keywords:** adolescents, mental health promotion, mental well-being, PRISMA, teenagers

## Abstract

Background: Research on youth positive mental health (PMH) lacks comprehensiveness. We reviewed literature to (i) identify and understand concepts related to youth PMH and (ii) to summarize their definitions under broad conceptual themes. Method: We conducted a systematic review using PRISMA methodology. The protocol was registered in PROSPERO database (ID:CRD42020203712). Pubmed, Embase, PsycINFO, and OpenGrey databases were searched for publications that examined, assessed, explained, or defined PMH concepts in youth populations. Methods included searching, independent screening and review using pre-determined inclusion and exclusion criteria, extraction, coding, and iterative thematic syntheses of literature. Results: Of 3427 unique records identified, 105 articles from 26 countries met review criteria. Qualitative analysis resulted in 22 broad themes of youth PMH. These included interpersonal relationships (interpersonal competence, school connectedness, etc.), positive emotions (feel and create pleasant emotions, gratitude, etc.), self-efficacy (strengths, human agency, etc.), life satisfaction (global assessment of one’s life), and personal growth (goal achievement, life aspirations, etc.). Five novel concepts related to youth PMH were identified. Conclusions: Our review summarized and operationalized multiple concepts of youth PMH for applications in research, evaluation, and public health improvement.

## 1. Introduction

Youth mental health is an area of focus in educational, epidemiological, clinical, and public-health research. In the mid-twentieth century, mental well-being was first conceptualized by Maslow from the perspective of a hierarchical pyramid of basic human-needs’ satisfaction and motivation to reach one’s full potential [1]. Maslow emphasized that meeting the psychological needs of love and self-esteem was critical for achieving human potential or self-actualization. This theory has often been applied to understand the relevance of healthy relationships, independence, and self-respect to youth mental health. However, a criticism has been that the model does not distinguish between the needs concerning negative and positive aspects of mental health [2]. A decade later, Marie Jahoda coined the term ‘positive mental health’ (PMH) and proposed six pillars of mental health relating to autonomy, attitude towards self, growth and self-actualization, integration, environmental mastery, and perception of reality [3]. Mental health and well-being were since described as ‘subjective’ (relating to the pursuit of pleasure and happiness) or ‘psychological’ (referring to one’s abilities and outlooks towards achieving one’s full potential) [4]. However, much of this work has focused on adult populations, and the application and comprehensiveness among youth are limited. From a developmental perspective, youth is a formative phase for self-identity, orientation, and growth. The psychological needs for well-being at this age are believed to be diverse and volatile compared with adults [5]. Hence, gaining an in-depth understanding of the components of PMH specific to youth is necessary for appropriate assessment and intervention.

This systematic review of the literature was conducted to: (i) identify and understand the concepts relevant to youth PMH, and (ii) to summarize their definitions within broader conceptual themes. 

## 2. Materials and Methods

### 2.1. Literature Search

A systematic literature search was conducted as per the PRISMA guidelines to identify the concepts used to explain, describe, and assess youth PMH and well-being. Primary searches were conducted for articles published between 1 January 1999 to 31 March 2019 in four databases–Pubmed, PsycINFO, Embase, and OpenGrey. The search strategy and search terms were developed after consulting a librarian and experienced mental health researchers (Table 1; Appendix A). The search terms included “PMH” and ten commonly used terms and theories [6]. Attempts to access the unavailable full texts were made through emails to their corresponding authors. 

### 2.2. Screening and Selection Criteria

The two-stage review protocol included eligibility assessment at the abstract level followed by full-text reviews. Each abstract was independently screened by at least two reviewers to assess their eligibility. The publications that examined mental health, well-being, and related concepts among youth were considered. Articles were included if they met the following criteria: (i) conceptualized mental health and/or components as a positive or beneficial state, trait, or construct. For example, studies evaluating happiness or self-esteem were included, while those solely assessing depression or neuroticism were excluded. The articles focusing on the development of measures were included if they provided operational definitions for the PMH constructs, factors, or subscales. (ii) study population aged between 12 to 35 years. This range was selected to include the youth age range of 15 to 24 years as defined by the United Nations [7]. We also included those below 15 and above 24 years to include studies among early teens and young adults to meet the wider youth age range adopted in Singapore [8]. The general principle for screening records was to be inclusive. In the instances where the study participants’ age was not specified, other characteristics such as mean/median age, class grade, or undergraduate status were considered, and studies were included if these characteristics were indicative of a youth population. 

Records were excluded for the following reasons: (i) not in English; (ii) focus on sociodemographic or biological/physiological factors (rather than concepts) related to PMH, for example, effect of time, age, or gender on mental health or biological factors; (iii) focused solely on negative aspects of mental health; (iv) no relevant focus on mental health; (v) editorials, books or book reviews; or (vi) intervention studies. Articles describing more than one study were included as long as at least one met the above criteria. The concordance in selection decisions between the reviewers was 76%, and of the remaining 24%, decisions were resolved by consensus discussions (116 articles) and/or by a third reviewer (12 articles). The final eligibility was determined by the first author for all of the selected and uncertain studies based on a full-text review. The risk of bias assessment was not performed as this review aimed to identify the definitions for youth PMH concepts, and there are no recommended quality assessment criteria for construct reviews [9].

### 2.3. Data Extraction 

The first author reviewed the selected full-text articles to extract passages and content with relevant definitions of PMH concepts used in youth populations. All of the sections in the articles were reviewed to record how the studied concepts were operationalized using content analysis [10] to capture definitions and/or descriptions of constructs, concepts, and subcomponents of youth PMH. Where authors used a priori definitions based on literature, theories, or measures, those best meeting the study aims were extracted. For the articles reporting results from more than one population, content specific to youth populations were recorded. Subsequently, the type and depth of information extracted was discussed between at least two authors to determine their relevance to the review. A subset of the selected full texts (20%) was also examined independently by another reviewer to ensure the appropriateness and relevance of the extracted data. 

### 2.4. Knowledge Synthesis

Text passages with descriptions and/or definitions were qualitatively analyzed using interpretative methods [11]. The first author (JAV) used open coding to generate operational definitions of the PMH concepts. Each definition was assigned a single open code to represent the most salient features of the text passage while being mindful of the contexts and novel ideas they presented. Where the concepts were defined using short phrases or definitions, the original text was adopted for the open coding. Given the wide array of definitions identified, the three reviewers (JAV, ES, SC) then examined similarities in the concepts and definitions to categorize them into broader themes around youth PMH. Each open code was allocated and classified within a single theme. The final thematic structure, nomenclature, and definitions were then discussed among the four reviewers (JAV, ES, SC, RS) to achieve consensus and refinement. Sensitivity analysis was performed to assess the impact of including studies with populations with ages outside of the UN definition of youth (i.e., 15 to 24 years) on the study findings. 

## 3. Results

The PRISMA flow chart of the search results is shown in Figure 1. A total of 3427 unique records were identified. Of these, 3085 records were excluded after the screening of abstracts, and 237 were excluded after eligibility assessment of full-text articles, resulting in 105 articles for review. 

### 3.1. Study Characteristics

The characteristics of the included studies are presented in Appendix A. Of these, 38 studies were conducted in teens and adolescents (below 19 years only), 2 in young adults past teenage years, and the rest of the studies were conducted in youth populations in general (across 12–35 years). Most of the studies (*n* = 88, 83.8%) included populations within the age range of 15 to 24 years, while 17 (16.2%) covered a wider age range of around 12 to 35 years. The studies were conducted in 26 countries; most originated in the United States of America (USA) (*n* = 36, 34.3%), followed by China/Hong Kong (*n* = 7, 6.7%) and Canada (*n* = 6, 5.7%). In addition, 10 (9.5%) were conducted internationally in populations of European, Latin American, or Asian origin from multiple countries. The study sample was obtained from academic institutes (schools, colleges, or universities) for 86 (81.9%) studies, while 19 studies used community samples. Three studies covered homeless adolescents [12], street kids [13], or ‘at-risk’ youth [14].

### 3.2. Core concepts of Youth PMH

Data extraction resulted in 469 text passages, some with synonymous and overlapping definitions. None of the articles in our review provided a multi-dimensional, comprehensive description or definition of youth-centric PMH or mental well-being. Therefore, individual concepts described as part of, relevant to, or representing youth mental health and well-being were categorized through inductive qualitative analysis of the text passages. Our results are anchored on these themes irrespective of how the authors had classified or labelled them. 

### 3.3. Conceptual Themes and Components of Youth PMH

The qualitative analysis resulted in 22 broad conceptual themes. Interpersonal relationship domains (e.g., positive relationships, interpersonal competence, and school connectedness) (16.1%); the experience of positive emotions/affect (e.g., feel and create pleasant emotions and gratitude) (10.1%); self-efficacy/competence (8.7%) (e.g., strengths, human agency, and perceptions of efficacy); life satisfaction (in the context of their overall life, friendship or school) (8.4%); and goal attainment/personal growth (e.g., achievement goal orientations, personal growth initiative, life aspirations) (8.2%) were the most commonly recurring themes. Three aspects—‘need satisfaction’, ‘psychological well-being’ and ‘subjective well-being’—are based on established theories [1,15,16] and not described in detail in our results. Five concepts that were not previously incorporated in PMH frameworks–‘faith’, ‘mindfulness’, ‘positivity’, ‘self-love’, and ‘vitality’–were identified and are hence discussed below as novel conceptualizations of PMH. The descriptions for these concepts are presented in the subsequent sections, and their simplified operational definitions are listed in Table 2. Due to overlaps among the concepts, the sequence of their descriptions is largely based on thematic similarities.

**Autonomy:** Autonomy was described as an “ability to act in ways that meet one’s choices” [17]. It is a feeling or sense of independence [18], volition [19], and being in command [20] of one’s personal choices and behaviors [19], school [21], prosocial activities [22], and religious/spiritual world-view [23]. At its core is the independent, self-generated, self-directed decision-making [24,25,26,27,28] and regulation of one’s actions [29] that originate from one’s self [22,30] and gives one a perception of being in command to choose [31,32].

**Identity/self-concept:** Identity or self-concept is defined as the autonomous or controlled orientation towards identity formation, commitment, and connection [25,33,34]. It represents a set of perceptions about oneself, based on personal assessment, evaluation of significant others [35], and exploration through personal experiences [33]. It involves the overall evaluation of personal traits and skills [36], self-beliefs about oneself and one’s identity [37], as well as self-evaluation of the present and desired identity [31]. Simply put, it is knowing who one is and what one is capable of achieving. The process of knowing, accepting, internalizing, and asserting self-identify through one’s personal, moral, and religious values [12] relates to self-concept. Interpersonal influences on these constructs result from interactions with important figures in adolescence, such as parents/peers [33], and can result in the evaluation of self-concept as independent of others’ opinions and help [12,26] or an interdependent appraisal [38] of how one’s sense of identity links with others in the society [39]. The construct also covers collective identity formed through racial, ethnic, or cultural affiliation [40]. Two studies also applied academic self-identify as students’ perception about themselves in the academic, moral, religious, and psychological contexts [41], and their perceived competence in academic areas [42]. 

**Self-esteem:** Self-esteem is defined as the positive evaluation of one’s “abilities, talents, close relationships” [43], good qualities and values [44], and self-identity [45]. It also relates to feelings of self-acceptance, goodness, and self-respect [46] and the evaluation of self-worth and adequacy [42,46,47] in academic and non-academic areas [48]. A broader definition is applied to explain “collective self-esteem” as the positive evaluation of one’s self-worth from being part of a group [43]. 

**Locus of control:** Five studies assessed the salience of control factors on PMH. Internal locus of control relates to one’s perceived sense of control and command over one’s life circumstances [49], future, and life goals [50]. It is the capacity to control thoughts, feelings, impulses, and behaviors [12,47]. Locus of control relates to one’s belief that one can determine the outcomes of one’s circumstances. This ‘sense’ can be derived from perceived self-directed decision-making (autonomy), one’s orientation towards oneself and the capacities (self-concept), and the positive self-evaluation of one’s attributes and circumstances (self-esteem). 

**Goal attainment/Personal growth:** The concept of goal attainment and personal growth covers one’s capacity, ability, and attitude [29] for intentional and active desire to grow in self-determined areas [49,51], life goals, and values [37], and having feelings of continued development [18] for realizing one’s full potential [25,52] and becoming one’s ideal self [45]. It relates to one’s ability to reach one’s goals despite challenges [53] by determining the best ways to achieve goals and feel competent [21,33,54]. It involves a cognitive process of subjective appraisal of one’s capacity, motivation, and plans [55] and self-direction [56] to set and meet long-term goals [34,51]. Goal attainment is marked by purposeful efforts [25] for competence, self-improvement, mastery [57,58], personal development, interpersonal connection, prestige, recognition, and spiritual goals [59,60]. A sense of achievement and motivation [57,61,62] is evident in the process of goal attainment and personal growth. Among the youth, this sense of achievement is often derived from school, leisure, personal appearance, and behaviors [62,63]. Motivations to set and achieve goals related to the desire for more [64], the expectation of success [42], tangible benefits to self and social standing [54] for themselves and others, such as parents [65].

**Meaning and purpose in life:** This concept refers to the feeling that one’s life is valuable and has a purpose as well as possessing that existential sense-making [20] in the “grand scheme of things” [66]. It also relates to having a drive for a manageable and meaningful life [34], having life goals and strategies to achieve them [18], being committed to personal values [25] and applying personal strengths to “belong to and serve” something bigger than oneself [62,67]. Life’s meaning is often derived from feelings of self-worth, self-efficacy, accomplishments, independence, and positive emotions [68] or through meaningful goals [69], religious beliefs, moral values, and contributing to the community [70]. For students, meaning is associated with the purpose and value attributed to achieving academic goals [71] and appraising academic tasks as important and meaningful [72]. 

**Engagement:** Engagement is generally described as a state of “psychological connection” [62] and continual engrossment, commitment, and absorption in tasks [73], places, motivating challenges [56] or life activities [53]. It also relates to one’s personal interest [74], perceived level of skill to face challenges [75], and focus on task performance [38]. In academic settings, it relates to behavioral and emotional engagement [21,76] and commitment to academic tasks [72]. Engagement is also envisioned as a group-level behavioral or cognitive commitment to social groups [77] and religious beliefs and practices [65]; often associated with being engaged in caring for others and feeling fulfilled through giving back to the community [69]. 

**Motivation:** Motivation is defined as possessing autonomous and self-determined reasons to pursue actions that have instrumental value, importance, or benefits attached to them [13] in the form of personal conquests [22,42,78]. It also relates to orientation towards prosocial activities that give happiness and satisfaction through concern for the welfare of others [60,79]. It is closely related to themes of goal attainment, self-efficacy, engagement, and positive emotions

**Resilience and coping:** Resilience is the ability to cope with, manage reactions to and recover from stress, risk, adversity, or change [29,76,80,81,82,83] and overcome barriers in goal pathways [84]. This process of recovering from stress involves relying on emotional instincts to determine coping actions and managing stress-generated emotions [39,85] and problem-solving through understanding, creativity, flexibility, and optimism [83,86]. In collectivist populations, coping also covers the ability to adopt responses and mechanisms mindful of and grounded in cultural/ racial norms—a practice referred to as collective coping [40].

**Self-efficacy and competence:** Self-efficacy relates to possessing the skills to regulate behavior, exploit opportunities, and navigate daily life [18,25,87] and the propensity to seek, accept, and adapt to unfamiliar experiences [50]. Self-efficacy is defined as the conscious appraisal of personal thought [14] and the perceived effectiveness of self-capacity. At the intrapersonal-level, it also covers the ability to generate and select coping responses [85], regulate negative emotions such as anger and rumination in stressful situations [44,86], and resolve problems and complete tasks effectively [42]. Academic self-efficacy involves perceiving one’s academic behavior as meeting the demands at school [72] and having confidence in completing academic tasks [61]. Inter-personal self-efficacy relates to the capacity for effective interaction through open communication, positive relationships, and conflict management [44]. Self-efficacy refers to the ability to use internal resources effectively to regulate emotions and behaviors [88], interactions with others and one’s environment to achieve the desired, and prevent the undesired, outcomes [18,19,30]. It also relates to perceived effectiveness in determining one’s religious/spiritual world-view [20], the experience of mastery [27], and perceived superiority in undertaking specific academic and non-academic tasks compared to others [42]. 

**Experience of positive emotions or affect:** A total of 42 studies covered this concept, which refers to one’s propensity to experience, feel, and navigate positive emotions (e.g., kindness, gratefulness, joy, content, and cheerfulness) and affect (e.g., pride, interest, excitement) [61,62,89], and gain a pleasant life [67,73]. Such experiences emerge from one’s cognitive assessment of one’s overall emotional reaction to life events [35,76,88], such as change and receipt of benefits [27,90]. PMH is an accumulation of such experiences [91] that result from one’s ability to monitor and regulate one’s emotions and behaviors [92], to gain more positive and fewer negative experiences [20,44]. Positive emotional experiences occur when one is mindful [93] and grateful for one’s surroundings [39,89], able to feel safe, empathetic, altruistic, generous, and compassionate about helping others and has a sense of connection and friendship [38,47,56]. In young students, PMH also results from their positive emotional and cognitive experiences from academic task engagement [72]. However, one qualitative study in Sweden associated positive emotions with ‘disengagement’, whereby students expressed feelings of independence and positive emotions by asserting their unwillingness to be involved in school tasks [63].

**Happiness:** Happiness relates to a positive cognitive assessment of one’s life [94] and in general is regarded as a good [95] or inner emotional feeling [96]. It is defined as a state of positive mood, contentment, and gratitude [53,65], determined by the intensity and frequency of positive emotions arising from relationships, accomplishments, and sense of direction [68,70], and associated with pleasure, engagement, and meaning [56]. As described by a group of homeless youth in USA, it is a feeling that “one is doing well” in life [12]. From a collective or community-oriented perspective, it is defined as a sense of enjoyment that originates through spending quality time with friends and family [97] and having a sense of harmony between one’s self, relationships, and social groups [39]. There are subtle distinctions between happiness and the ‘experience of positive emotions/affect’. Rather than being regarded as a positive emotion/affect, happiness is believed to aggregate multiple and frequent positive emotions [98]. It is more stable than intense emotions, such as joy or bliss [98,99], yet less stable than trait-based emotions such as being calm and peaceful [100]. Therefore, while it is possible that positive emotions contribute to one’s happiness, it is not necessary that one experiences happiness in the same way as positive emotions. Likewise, in relation to similarities with ‘life satisfaction’, contentment is a momentary emotional state resulting in happiness, while life satisfaction is a deeper cognitive judgement on one’s life [84]. For example, one can be very happy while spending time with friends yet not be fully satisfied with one’s life.

**Interpersonal relationship domains:** This was the most extensively covered theme with multi-faceted content covered in 67 studies. The general premise referred to connectedness/relatedness, defined as the tendency to seek and maintain dependable and close interpersonal connections and form meaningful, positive, and engaging relationships with others [23,31,101]. Interpersonal relationship domains also related to one’s strengths and abilities to empathize and seek, form, and maintain interpersonal relationships [29,102] and to empathize or relate to emotional experiences of self and others [87]. Other actions related to partaking in acts that benefit others [22,70], searching and forming significant connections with others [31], showing appreciation, cheering others, being trustworthy and helping others [103]. Several references related to the feeling, perception or tendency to be connected with others or integrated with desired groups [20,30,62,64,65,103], being cared for [53,62,72], loved [104], safety [65,105], and support [12,21,30,35,53,62,76,103,106]. Support systems among youth covered significant others [30], close relations [25,103], social networks [35], parents or non-related adults in one’s neighborhood [107] and teachers [76]. Connectedness with a community relates to one’s self-determined and positive orientation towards society and others [29] and being able to help and fulfil obligations, derive pleasure in other’s happiness, and in some instances, also relates to beliefs in religion or God [65]. Within an academic setting, positive inter-personal relationships referred to “the extent to which students feel accepted, respected, included, and supported in the school environment” by teachers and peers, both personally and academically [21,76,106]. Other aspects included feeling “cared for by and relating well to others at school” [72], being committed and comfortable at school by adapting to the school system [21], perceived adequacy of support from teachers [76], social competence [24], and perceived prosocial behavior towards others in school [72]. 

**Life satisfaction:** Life satisfaction is widely defined as the cognitive appraisal of one’s life based on a range of individually determined milestones [35,44,82,84], subjective or cognitive evaluation of one’s life’s quality [39,108] or one’s own internal standards which may or may not be recognized by others [109]. This evaluation can be global or an assessment of one’s life as a whole [23,32,36,40,61,76,92,107,109,110], or domain-specific [73] relating to different areas of life such as friendships and academic satisfaction from important features of one’s academic life [39,61]. 

**Need satisfaction, and psychological and subjective well-being:** Need satisfaction relates to the degree to which one feels supported for one’s basic psychological needs of autonomy, competence, and relatedness. In school settings, it refers to students’ need satisfaction and frustration regarding their studies [58]. In the context of relationships, need satisfaction is achieved when one feels connected to and understood by others [19,105]. Well-being was assessed in 17 studies using established questionnaires to assess subjective [22,70,75,111,112,113,114] or psychological well-being [13,34,39,40,45,49,81,83,115,116,117]. Generally, subjective well-being relates to happiness, being satisfied with life and the presence of positive and negative affect. In contrast, psychological well-being is conceptualized as the ability to forge positive relationships, feel good about oneself, function effectively, experience positive emotions, affection, confidence, interest and engagement, and having a purpose in life. 

**Novel concepts:** Five concepts related to youth PMH—‘mindfulness’, ‘faith’, ‘positivity’, ‘self-love’ and ‘vitality’—emerged from some of the studies among the youths. These were conducted almost exclusively in Western regions and included: (i) Mindfulness, which is defined as the awareness, attention, and positive approach to present experiences [118]. It also relates to self-regulated attention, openness, and acceptance of immediate experiences [119]. (ii) Faith, which relates to the felt or perceived connection with one’s religious or spiritual world-view, divine being, or reality [23]. Combined together, they cover the subjective feeling of interest and deep understanding of life meaning [81] and the “feelings, thoughts, and experiences that arise from a search for the sacred or transcendent” [20]. Faith is believed to enable the internalization, expression, and having open dialogues about life experiences and meaning through religion [23,83,120], generating a culturally congruent meaning of self, others, and the society [69] and thereby nurturing a sense of autonomy, competence, life satisfaction, and relatedness [23,79]. (iii) Positivity, which relates to being open to new experiences [75], adopting a favorable, hopeful, and confident view about the future, and discounting adverse events as being temporary [53]. It also relates to having optimistic, positive expectations about the future and harboring positive views of self, others, future, and life [44,121]. (iv) Self-love, which is one’s evaluative process of understanding and accepting oneself and one’s strengths, flaws [45], and past [25]. It relates to one’s attitude towards who and what one is and involves possessing a caring attitude towards self while in adversity [114]. (v) Vitality, which is considered a positive emotion [30] and having “positive energy, purposefulness, and a sense of aliveness, and zest” [60]. Such energy is necessary to engage in activities and pursuits that produce happiness [50]. Vitality also relates to the positive energy in emotions, thoughts, behaviors, and outlook [88].

### 3.4. Age and PMH Concepts 

Sensitivity analysis performed to assess the impact of the inclusion of studies outside the UN-defined youth age range did not indicate any major variation in the concept distribution or content, except for more information related to interpersonal relationship domains (Table 3). The thematic content in studies conducted solely among teenagers/adolescents varied as compared with youth in general. Among teenagers, there was a greater emphasis on ‘Engagement’, ‘Happiness’, ‘Interpersonal relationships’, ‘Life satisfaction’, ‘Locus of control’, ‘Self-efficacy’, ‘Subjective well-being’, and ‘Vitality’, and less on ‘Autonomy’, ‘Goal attainment/ Personal growth’, ‘Motivation’, ‘Faith’, ‘Resilience/coping’, and ‘Self-love’ (Table 3). However, given that only two studies were conducted solely with non-teenagers, we were unable to investigate such differences further. 

### 3.5. Geographic Location and PMH

More studies were conducted in North America (*n* = 42), with a majority carried out in the USA (*n* = 36) and the rest in Canada. The studies from Europe formed the second largest group with studies across 12 countries, 5 of which were conducted in Spain. Sixteen studies were carried out in Asia; the majority (*n* = 7) were from China/Hong Kong. The international studies were conducted in Western and Asian settings. Few differences were observed across regions in terms of the concepts studied (Table 4). 

## 4. Discussion

Our review highlights the extensive diversity in the composition of youth PMH. We combined information from 105 studies assessing the links between and the structure of the concepts relating to emotional and psychosocial mental health of adolescents, teenagers, and young adults. This review showed that a comprehensive, youth-focused framework for youth PMH is not yet available. We also found that definitions of PMH concepts are not standardized, and vary widely concerning the focus on cognitive, emotional, functional, and behavioral standpoints, often resulting in overlap between concepts and permutations. Our results thus suggest that youth PMH is a multi-dimensional, multi-factorial construct. Our results are supported by proponents of multi-dimensional models of person-reported outcome measures that enable the comprehensive scanning and assessment of psychosocial and behavioral constructs [122,123]. Most of the conceptual themes identified and summarized in our review were included in existing models of mental health and well-being [4]. 

However, these also included five novel construcsts—mindfulness, positivity, faith, self-love, and vitality—that are under-studied among youths and do not feature distinctly within any of the frameworks. As part of our review, we defined these constructs from a lifestyle and mindset standpoint as a psyche that could be inculcated in all aspects of one’s being, thereby anchoring them as essential components of youth PMH. Although their heuristic, conceptual, and linguistic relevance to today’s youth is palpable, how strongly they explain the construct of youth PMH can only be assessed through further research and empirical evidence. Nevertheless, there is support for their inclusion in assessments of mental health. Mindfulness is increasingly regarded as a lifestyle choice from both a clinical and developmental perspective. A recent review found that routine mindfulness was associated with beneficial personality and mindset changes in adolescents and effectiveness in elevating mood and preventing excessive alcohol consumption and eating disorders [124]. Positivity is distinct from optimism, in that optimism is considered as an outlook during stressful events, whereas positivity applies to “agency, insights, capabilities and contributions” during daily routines [125]. 

Faith has long been associated with mental health benefits [126], but there is no consensus on whether it is an aspect of mental health or an associated factor. Religious and spiritual beliefs and practices are believed to directly benefit mental health by increasing self-confidence, hope, and a sense of purpose, and indirectly through improved moral values, healthy behaviors, and a social network and support [127,128]. Specifically, among the youth, religious and spiritual beliefs are believed to influence their cognitive and emotional orientation away from self and materialism towards prosocial behaviors and “authentic concerns about the world” [129]. Such tendencies provide meaning to their lives and motivate them to live in a mindful and considerate manner. Thus, there is adequate support for assessing religious or spiritual faith within youth mental health.

Self-love has conceptual similarities with self-compassion and mindfulness-based healing that have proven beneficial to adolescent and youth mental health [130]. Self-love, as we define it, focuses on the ‘self’ over the ‘present’ (as implicit in mindfulness) and disregards the adversity-situational context of self-compassion (e.g., being compassionate in adversity) and applying self-love to the everyday routine of youth. Enthusiasm, openness, and extraversion that have been extensively studied in adolescent and youth mental health have similarities with the concept of ‘Vitality’ [131]. These concepts also feature as beneficial traits and building blocks within some of other themes we identified. For example, feeling energetic is a positive emotion, while having a sense of direction and feeling of engagement in goals are featured as ‘goal attainment’ and ‘engagement’. Vitality is an aggregate outcome of these emotions and can provide a quick assessment of ‘zest’ in youth. 

Our findings have implications for research and public health. First, we highlight the need to have a comprehensive framework for youth PMH. This would require knowledge of the critical aspects of youth mental health and the factors that influence them within the prevailing social and cultural contexts. Gaining an understanding of the immediate and long-term impacts of common experiences in youth is equally relevant. In addition, engaging youth as the agents of change in this process of discovery would yield effective identification of youth PMH foci. This review can serve as a starting point in the identification of youth-relevant components of mental health. While we were able to indicate which areas can be targeted for developing interventions by operationalizing youth PMH as a complex construct, a challenge arises in determining the boundaries of these constructs due to overlapping definitions. This hampers a clear evaluation of the effectiveness of focused interventions for mental health improvement and promotion. Second, there are scientific and practical benefits to defining youth PMH concepts in terms of the development of measures and the identification of health promotion priorities and interventions. Thus, having universally accepted and operationalizable definitions for the youth PMH concept is crucial. 

### Strengths and Limitations

Our systematic review and qualitative synthesis summarized the growing literature on youth positive mental health and well-being. We used a comprehensive range of search terms and adopted a structured approach to knowledge synthesis. Additionally, we included peer-reviewed dissertations that allowed inclusion of unpublished literature. However, our analysis may have been affected by the search period and the age limits imposed in defining the youth populations. Updates to this review should, therefore, consider including wider search strategies and criteria. Another limitation was our approach to investigate relevant concepts within the last two decades. We did not consider their historical development or the context within which each was established, which, thus, may have biased the results towards recency. Despite these limitations, this systematic review covered a large volume of literature analyzed using rich textual review, and presents comprehensive, up-to-date, and as-yet unavailable information on what constitutes youth PMH. 

## 5. Conclusions

This review intended to comprehensively capture concepts of PMH that were considered by researchers as important to youth mental health, in order to summarize their operational definitions described in the studies included in the review. The findings are meant to help researchers, mental health professionals, policymakers, parents, and youth themselves to understand the multidimensional constructs underpinning youth mental health. The review can also help chose relevant concepts of youth positive mental health while designing and evaluating mental-health interventions. The review showed that youth positive mental health is a complex and multidimensional construct. There are extensive overlaps between the different components of youth PMH, thus, it is important to consider the multiple likely starting points while assessing or improving youth mental health. The results of the review can help to disentangle the relative effectiveness of different approaches to youth mental wellbeing and promote more effective and appropriate mental health promotion among the youths. Future consideration for research could include a review of literature based on the cultural and educational settings that youth are exposed to, to provide a more nuanced understanding of global environmental and public health impacts on youth mental health.

## Figures and Tables

**Figure 1 ijerph-19-11506-f001:**
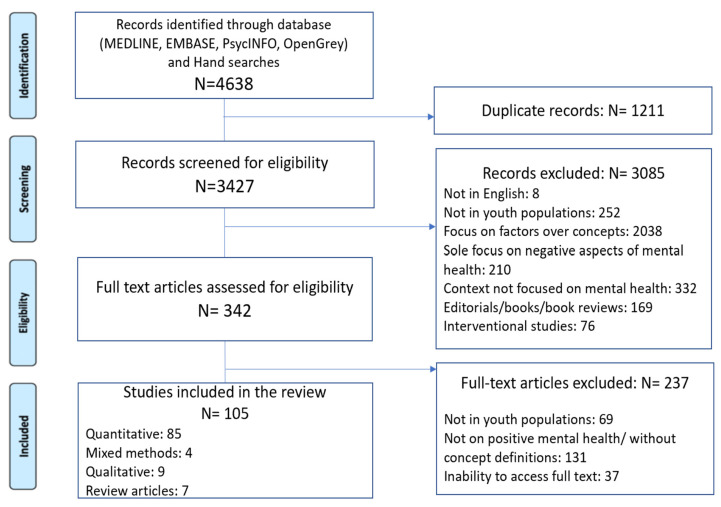
PRISMA flow chart of search results.

**Table 1 ijerph-19-11506-t001:** Broad search strategy to identify articles on positive mental health (PMH) in youth.

Part 1	Part 2	Part 3
**Rationale:** The aim of our review is to understand the conceptual framework of youth positive mental health	**Rationale:** The focus was on youth populations largely in the UN definition of youth (15–24 y).	**Rationale:** There is no universal definition of youth PMH. A range of terms are used to assess mental health and well-being in youth.
conceptual framework	Youth*	mental well-being
theory	Adolescent*	mental wellness
hypothesis	Teen*	positive mental health
theoretical framework	Young adult*	psychological well-being
concept	Young people	subjective well-being
measurement	Student*	self-determination theory
	Adolescence (MeSH)	hierarchy of needs
		happiness
		life satisfaction
		self-actualization
		big five personalit*

**Table 2 ijerph-19-11506-t002:** Summarized operational definitions of youth positive mental health concepts.

Concepts of PMH	Definitions
Autonomy	Being in command of one’s personal choices and behaviors
Engagement	State of continual engrossment, commitment and absorption
Experience of positive emotions/affect	Ability to experience, feel and create pleasant emotions
Faith	Felt or perceived connection with one’s religious and/or spiritual world-view
Goal attainment/ Personal growth	Active and continual desire to grow in self-determined areas
Happiness	Subjective state of positive mood and contentment
Identity/self-concept	Knowing and evaluating who one is and what one is capable of achieving
Interpersonal relationship	Ability and process of seeking, forming, and maintaining close relationships with others
Life satisfaction	Subjective assessment of how one’s life has turned out to date
Locus of control	Having a sense of control and command over their life
Meaning and purpose in life	Feeling that one’s life is valuable, worthy, and directed
Mindfulness	Having awareness, attention, and positive approach to present experiences
Motivation	Drive to pursue actions to reach goals for oneself and others
Need satisfaction	The degree to which one feels support for their basic psychological needs in various domains
Positivity	Possessing positive outlook and openness to new and unfamiliar experiences
Psychological well-being	Ability for having positive relationships, emotions, functioning, outlook and purpose in life
Resilience/coping	The process of recovering from stress through evaluation and coping
Self-efficacy/competence	Possessing skills and tendency to effectively reflect, seek, and modify situations in one’s favor
Self-esteem	Positive evaluation of one’s self-worth
Self-love	Gentle and caring attitude towards who and what one is and was
Subjective well-being	Experience and frequency of positive emotions and overall life satisfaction
Vitality	Having positive energy, resolve, and zest for life

**Table 3 ijerph-19-11506-t003:** Distribution of positive mental health concepts by age of the study population.

	Age-Related Differences				Sensitivity Analysis			
Conceptual Themes of PMH	Adolescents and Teenagers (Below 19 years) *N* = 38		Youth in General (12–35y) *N* = 67		All Full Texts (12–35y) *N* = 105		Excluding 17 Studies outside UN Youth Age Range *N* = 88	
	*n*	%	*n*	%	*n*	%	*n*	%
Autonomy	1	0.7	15	5.4	16	3.8	13	3.7
Engagement	7	5.0	7	2.5	14	3.4	14	4.0
Experience of positive emotions/affect	14	10.0	28	10.1	42	10.1	35	10.1
Faith	0	0.0	16	5.8	16	3.8	16	4.6
Goal attainment/Personal growth	8	5.7	26	9.4	34	8.2	28	8.1
Happiness	6	4.3	5	1.8	11	2.6	9	2.6
Identity/self-concept	8	5.7	16	5.8	24	5.8	22	6.3
Interpersonal relationship	30	21.4	37	13.4	67	16.1	47	13.5
Life satisfaction	14	10.0	21	7.6	35	8.4	31	8.9
Locus of control	3	2.1	2	0.7	5	1.2	3	0.9
Meaning and purpose in life	5	3.6	9	3.2	14	3.4	14	4.0
Mindfulness	1	0.7	1	0.4	2	0.5	2	0.6
Motivation	3	2.1	12	4.3	15	3.6	12	3.5
Need satisfaction	1	0.7	7	2.5	8	1.9	5	1.4
Positivity	4	2.9	9	3.2	13	3.1	9	2.6
Psychological well-being	3	2.1	8	2.9	11	2.6	10	2.9
Resilience/coping	2	1.4	11	4.0	13	3.1	12	3.5
Self-efficacy/competence	15	10.7	21	7.6	36	8.7	31	9.0
Self-esteem	7	5.0	14	5.1	21	5.0	19	5.5
Self-love	0	0.0	5	1.8	5	1.2	4	1.2
Subjective well-being	3	2.1	3	1.1	6	1.4	5	1.4
Vitality	5	3.6	4	1.4	9	2.2	6	1.7
Total	140		277		417		347	

‘*n*’—refers to the number of concepts; a majority of studied covered more than one concept.

**Table 4 ijerph-19-11506-t004:** Distribution of positive mental health concepts by region were studies were undertaken.

	Asia *N* = 16	Australia *N* = 4	Europe *N* = 22	North America *N* = 42	South Africa *N* = 1	South America *N* = 3	Inter-National *N* = 17
	*n*	*n*	*n*	*n*	*n*	*n*	*n*
Autonomy	-	-	3	9	-	-	4
Engagement	1	1	3	2	-	1	6
Experience of positive emotions/affect	4	2	10	12	1	1	12
Faith	1	1	3	10	-	-	1
Goal attainment/Personal growth	5	1	8	12	2	-	6
Happiness	5	-	1	2	-	-	3
Identity/self-concept	-	1	4	13	-	-	6
Interpersonal relationships	10	2	12	40	-	2	1
Life satisfaction	5	1	6	14	-	3	6
Locus of control	-	-	1	3	-	-	1
Meaning in life	3	1	3	4	-	2	1
Mindfulness	-	-	2	-	-	-	-
Motivation	1	-	4	9	-	-	1
Need satisfaction	2	-	-	6	-	-	-
Positivity	1	1	5	4	-	-	2
Psychological well-being	1	1	2	6	-	-	1
Resilience/coping	2	-	3	7	-	-	1
Self-efficacy/competence	4	1	7	16	-	1	7
Self-esteem	5	-	6	6	-	1	3
Self-love	-	-	1	4	-	-	-
Subjective well-being	2	-	1	2	-	1	-
Vitality	-	-	1	8	-	-	-
Total	52	13	86	189	3	12	62

‘*n*’—refers to the number of concepts; most of the studies covered more than one concept.

## Data Availability

Not applicable.

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
