# Peer review of "Youth Positive Mental Health Concepts and Definitions: A Systematic Review and Qualitative Synthesis"

_ijerph, 2022, doi:10.3390/ijerph191811506_

Round 1

Reviewer 1 Report

Sound qualitative inquiry with well-stated biases and limitations duly noted. Opens the door for furthering tacit, implicit, and explicit knowledge production about concepts and definitions in positive mental health in youth. Future consideration in determining study selection criteria ought to include references about cultural imperialism within colonized Western education and research. This includes pedagogical  possibilities about integrative thinking relative to how globalized western educational ideologies impact early youth development. A more metatheoretical environmental psychology focus would shape the primary prevention of psychopathologies and the global environmental public health impact of both youth and adult neuroses.

Author Response

We would like to thank the reviewer for highlighting future directions for research from our work. We have added a sentence in the discussion based on this feedback.

Reviewer 2 Report

A very interesting and well conducted study using a systematic approach to bring together themes. Well done

Author Response

We would like to thank the reviewer for the kind words of encouragement.

Reviewer 3 Report

The manuscript represented a systematic review using PRISMA methodology and a qualitative synthesis of terms and definitions related to positive mental health in youth. To group definitions under overarching conceptual themes, the authors searched PubMed, Embase, PsycINFO, and OpenGrey, four well-known databases, for publications that examined, assessed, explained, or defined positive mental health concepts in youth populations.

After conducting a literature review of articles published between 1999 and 2019, the authors discovered that there were 4638 publications that satisfied the search criteria. In the end, after applying the inclusion criteria in the study, there were 105 articles to be reviewed.

I recommend that the authors double-check the number of excluded articles. Are the 76 interventional studies included in the excluded records mentioned above (not in English, not in youth, etc.)?

The article is very well documented, and the topic addressed is current. The authors conducted a comprehensive analysis of the literature and made a rigorous selection of the bibliography in order to identify all PMH concepts. The result of their work resulted in the identification of 22 concepts, of which 5 were not previously incorporated into the PMH framework: faith, mindfulness, self-love, and vitality.

I appreciate that the authors' research will undoubtedly be useful to mental health professionals, researchers, policymakers, parents, and youth themselves in understanding the multidimensional constructs that underpin youth mental health, and thus I strongly recommend that this article be published. 

I have also included the feedback in the attached paper.

Author Response

Thank you for your kind review and identifying an error in the PRISMA chart. We have checked our articles and noted that interventional studies were double counted as also those with a focus on factors over concepts. We apologize for this oversight. We have corrected the numbers in the revised figure.